# Advances in Highly Ductile Concrete Research

**DOI:** 10.3390/ma17184596

**Published:** 2024-09-19

**Authors:** Jingjing He, Zhibin Huang, Xuezhi Wang, Ming Xin, Yong Zhang, Haodan Lu

**Affiliations:** 1Power China Northwest Engineering Corporation Limited, Xi’an 710065, China; hejing_86@126.com (J.H.); 18066968193@163.com (Y.Z.); luhaodann@163.com (H.L.); 2School of Civil and Architectural Engineering, Liaoning University of Technology, Jinzhou 121001, China; xmmyemail@163.com

**Keywords:** high-ductility concrete, fiber properties, mechanical properties, wet and dry cycles, freeze–thaw cycles, salt erosion, durability

## Abstract

In recent years, high-ductility concrete (HDC) has gradually become popular in the construction industry because of its excellent ductility and crack resistance. Concrete itself is a kind of building material with poor tensile properties, and it is necessary to add a large number of steel bars to improve its tensile properties, which increases the construction cost of buildings. However, most of the research studies on high-ductility concrete are scattered. In this paper, the basic mechanical properties of high-ductility concrete and the effects of dry and wet cycles, freeze–thaw cycles, and salt erosion on the durability of high-ductility concrete are obtained by comprehensive analysis. The results show that the tensile properties of HDC can be significantly improved by adding appropriate fiber. When the volume fraction of steel fiber is 2.0%, the splitting tensile strength of concrete is increased by 98.3%. The crack width threshold of concrete chloride erosion is 55–80 μm, and when the crack width threshold is exceeded, the diffusion of CL-1 will be accelerated, and the HDC can control the crack within the threshold, thereby improving the durability of the concrete. Finally, the current research status of high-ductility concrete is analyzed, and the future development of high-ductility concrete is proposed.

## 1. Introduction

With the technological progress and transformation and upgrading in the field of construction, scholars are paying more and more attention to the research on concrete properties, especially on the mechanical properties as well as durability. It has been found that since the end of the last century to date, the area of China at risk for building durability problems has been as high as 2.34 billion m^2^ [1]. In addition, with China’s western development strategy, the pace of infrastructure construction continues to accelerate, while the concrete working environment is increasingly harsh, such as the Tibetan Plateau region of China, due to its average plateau altitude of 4000 m, resulting in a great difference between day and night temperatures, and in extreme cases the temperature is as low as −45 °C, which has caused a great impact on the concrete structure [2]. As a result, some buildings in special environments have emerged, with higher requirements for the toughness, crack resistance, and ductility of concrete, which has led to the research and development of highly ductile concrete.

High-ductility fiber concrete (HDC) is a material with high ductility, high crack resistance, and damage resistance proposed by Professor Victor C. Li [3] of the University of Michigan in the last century. Since then, the United States, Japan, the United Kingdom, and other countries have carried out systematic research on HDC and gradually applied it in practical engineering. In recent years, HDC has been gradually applied to concrete structural reinforcement, vibration damping, vibration mitigation, and beam–column node components [4]. Its fine aggregate particle size is due to the very-low-permeability cement or fly ash admixture used to meet the effective bonding with the concrete components, while its addition of good-durability polyvinyl alcohol polyethylene fibers can effectively control the cracks to carry out multiple cracking and result in a very small crack width. Usually, the largest crack is about 100 μm, with high tensile strength, extensibility, and energy consumption capacity [5]. HDC is a new type of concrete developed based on high-performance fiber-reinforced cementitious composite [6] (ECC), which adopts the optimal fiber admixture to make its tensile stress nearly 10 times higher than that of ordinary concrete. Compared with the brittle damage characteristics of ordinary concrete, high-ductility fiber concrete under tensile and shear loading embodies multi-crack development and strain-hardening characteristics, which significantly improves the ductility and damage-resistant capacity of the component, and has a wide range of application prospects in civil engineering reinforcement [7].

## 2. Basic Properties of Highly Ductile Concrete and Fibers

### 2.1. Basic Characteristics of HDC

HDC stands for engineered cementitious composite, which is a fiber concrete made by mixing cementitious materials, fine aggregates, water, fibers and other additives. It follows the principles of fine mechanics and fracture mechanics and considers fiber bridging. It exhibits significant strain-hardening characteristics under tensile conditions and high toughness under compressive conditions. Overcoming the brittleness of ordinary concrete, it can effectively improve the ductility of the structure; it has the characteristics of stopping cracks and increasing density, and it has good application effects in structural reinforcement and earthquake resistance. The studies on the effects of different types of concrete subjected to sulfate erosion and wet–dry cycles mainly focus on mechanical properties, transport properties, and mechanism analysis [8].

### 2.2. Different Fiber Properties

The current fibers often incorporated in concrete mainly include steel fibers and organic fibers, which are divided into polypropylene fiber (PPF), polyethanol fiber (PP), polyvinyl alcohol fiber (PVA), and polyethylene fiber (PE). The reinforcement mechanism of steel fibers, due to their high strength and modulus, is mainly through the bond between the fibers and the concrete matrix. When the concrete cracks, steel fibers can span the cracks and transfer the stress, thus organizing the generation and development of cracks, thus significantly improving the tensile strength and crack resistance of concrete. Organic fibers have a good flexibility and dispersion, presenting a three-dimensional chaotic distribution in the concrete, which can inhibit the generation of cracks, thus improving the durability and crack resistance of concrete.

Some scholars [9,10,11,12,13] have conducted research on the high-toughness PVA fiber concrete material compressive test, showing that the concrete does not crack as early, specimens without the chipping phenomenon occur, and after the destruction of the concrete, reflecting a good integrity, no obvious phenomenon of convexity and spalling is seen; the compressive damage of the concrete changes from brittle damage to ductile damage. As the length of the fibers added to the concrete gradually increases, the compressive strength of the concrete decreases. Fibers added to highly ductile concrete create bridging links within the matrix and are able to meet the reinforcement requirements. Jun Zhang et al. [14,15] conducted uniaxial tensile tests on PVA fiber concrete and found that the stress–strain relationship of highly ductile fiber concrete under tensile loading can be divided into three phases, which are the elastic rise phase, strain-hardening phase, and strain-softening phase. Compared with ordinary concrete, fiber concrete has a more stable falling stage. The mechanical properties of highly ductile concrete are continuously improved with the increase in the PVA fiber admixture. It has been found by Said et al. [16] that when the volume admixture of PVA fibers is 1.5%, the tensile effect of the fibers can be fully exploited, which significantly improves the ductile properties of HDC.

Wu Hazy et al. [17,18] conducted static tensile tests on PE fiber concrete. The study showed that regarding the tensile strength of the structure, the load-carrying capacity with the increase in the amount of fibers continues to improve; this is because the effective bridging between the fibers effectively inhibits the generation and development of cracks. The test proved that the moderate addition of fibrous materials can significantly improve the crack resistance and ductility of the components. Professor Yu et al. [19] of Tongji University developed an ultra-high-ductility concrete material (UHDC), which has both the tensile deformation capacity of steel and can meet the requirements of the concrete fluidity of the material (shown in Figure 1). Figure 1 shows the observed results of the digital image correlation technique for cracks in plain concrete beams and UHDC beams, and the comparison reveals that the flexural and shear cracks in UHDC beams are more intensive, and they exhibit a better ductility in shear as compared to plain concrete beams.

In the past, scholars mostly studied highly ductile concrete with PVA fibers, but UHDC uses PE fibers. PE fibers [20] have a higher strength and elasticity compared to PVA [21] fibers. In addition, PE fibers are hydrophobic compared to PVA fibers. The incorporation of PE fibers enhances the toughness of the concrete and changes the brittle characteristics of the concrete, making it more difficult to rupture under stress. Table 1 [22] demonstrates the common parameters of both fibers.

The addition of steel fibers to concrete can effectively improve the strain capacity and tensile properties of concrete. Song et al. [23] analyzed the mechanical properties of steel fiber-reinforced concrete, and found that when the volume fraction of steel fiber admixture was 1.5%, the fibers improved the compressive strength of concrete the most, which increased by 15.3% year-on-year. While its splitting tensile strength and modulus of rupture increased by 98.3% and 126.6% respectively at 2.0% volume admixture (shown in Table 2 and Table 3).

In summary, scholars have extensively studied the effect of different fibers incorporated into concrete properties, because PVA, PE, and steel fibers have a high strength, abrasion resistance, corrosion resistance, and good processability, which provides them with a wide range of prospects for application in the fields of civil engineering, construction, composite manufacturing, and environmental protection and governance. Although there are different fibers in the structure, performance differences, and other aspects of difference, the inclusion of fibers is shown to enhance the tensile and strain properties of concrete. In addition, the excellent performance of fibers provides strong support for innovation and development in various fields.

## 3. Mechanical Properties of HDC

### 3.1. Static Mechanical Properties of HDC

For the static mechanical properties of HDC, scholars have mostly focused on the study of several aspects such as the type of doped fibers, doping amount, size, shape, and mechanical properties under different environments, and the specific studies will be discussed in the following aspects.

#### 3.1.1. Effect of Different Fibers and Dosage on HDC Performance

Scholars have now generally concluded that the stability, damage resistance, and safety of highly ductile concrete are higher than that of ordinary Portland concrete (OPC) [24]. The reasons are as follows: (1) PVA and other fibers have a good ductility and good tensile properties, and their incorporation into the concrete improves the working performance of the concrete; (2) because of its performance regarding not being easy to crack, it is better than OPC in terms of damage resistance. Zhang et al. [25] conducted reinforcement tests on plain concrete using mineral admixtures such as silica fume and nano-silica in order to investigate the mechanical properties of multiscale mixed-fiber high-performance concrete. It was found that the silica fume admixture of 6% increased the compressive strength of concrete the most significantly. And the fiber test was carried out by mixing 20 mm-long steel fibers, 9 mm-long polyvinyl alcohol fibers, 12 mm-long basalt fibers, and 20 μm-long calcium carbonate whiskers in concrete. It was found that the mixing of the above fibers reduced the compressive strength of concrete to different degrees, but the fibers were able to improve the flexural strength of the concrete. It shows that when the fiber dosage is appropriate, the hybrid fiber has the best reinforcing effect on concrete and can achieve the best synergistic effect of fiber and concrete.

With the continuous development and improvement of technology, the application of 3D printing technology in highly ductile concrete has been substantially improved. Three-dimensional printing was first introduced by Joseph [26] in 1997. It was not until 2003 that Khoshnevis [27,28] realized the actual 3D printing of concrete using a new method called “contour fabrication”, which eventually evolved into extruded concrete printing [29]. Three-dimensional concrete printing (3DCP) has many advantages, such as being easy to operate, having fast construction, a light weight, and so on. Ye et al. [30] investigated the use of 3DCP as a raw material for UHDC to study ultra-high-ductility concrete with proper compatibility. From the effect of different fiber contents on the UHDC sum properties as well as the mechanical anisotropic properties, and in order to better analyze the anisotropic behavior of UHDC, Equation (2) was proposed based on Equation (1) proposed by Ma [31] et al.
(1)$$I_{a}=\sqrt{{\left(f_{X}-f_{MC}\right)}^{2}+{\left(f_{Y}-f_{MC}\right)}^{2}+{\left(f_{Z}-f_{MC}\right)}^{2}}/f_{MC}$$
(2)$$I_{Direction}=\sqrt{{\left(f_{X}-f_{Direction}\right)}^{2}+{\left(f_{Y}-f_{Direction}\right)}^{2}+{\left(f_{Z}-f_{Direction}\right)}^{2}}$$

Eventually it was found that 3D printing could be used to construct UHDCs. It was found that UHDCs with a fiber content of 1.5% had a better performance in all aspects and reduced costs in terms of materials.

Ye et al. [32] investigated the invention of a new ultra-highly ductile concrete (UHDC) printed from 3D concrete, which is modified by rubber and has a high ductility. The test results showed that the compressive, flexural, and shear strengths of the printed specimens were slightly lower than those of the molded specimens due to the higher torsional orientation of the fibers, but the opposite was true for the compressive strength. The deformability and energy dissipation in some directions of the printed UHDC were higher than those of the molded UHDC. In addition, the printed UHDCs were found to be less anisotropic in terms of flexural strength and exhibited significant anisotropy in terms of flexural deformation capacity, compression capacity, and flexural energy dissipation capacity. Li et al. [33] investigated the 3D concrete printing process (see Figure 2). From left to right, the concrete container is first connected to the drive motor, and then the concrete is transported through the delivery pipe to the pumping equipment for mixing. Finally, the printed sample is formed after the extrusion pump, and below the image there are photos showing the printed concrete structure and its internal characteristics, with gaps between the weak interface and the adjacent layers. It was found that the printed concrete showed anisotropy, which was verified by an ultrasonic pulse velocity test. The fiber orientation and fiber strengthening mechanism were analyzed by scanning electron microscopy.

The incorporation of coarse aggregates in highly ductile concrete is currently used as a means to reduce production costs and facilitate further engineering applications [34]. Xu et al. [35] prepared highly ductile concrete containing coarse aggregate and analyzed the effect of incorporating coarse aggregate and steel fibers on the mechanical properties of concrete and the synergistic effect of the two. It was found that the compressive and tensile strengths were increased at 0–28% of the coarse aggregate admixture, but when the coarse aggregate was increased to 38% of the admixture, the strengths began to decrease; the incorporation of coarse aggregate weakened the flexural properties of the high-performance concrete.

Shaped columns can effectively increase the use area of the interior and are economical and aesthetically pleasing, which meets the needs of people living in the house [36]. However, due to the irregularity of its cross-section, the concrete is more prone to cracking during stress, and the brittle characteristics are more obvious. It has been found that the addition of highly ductile concrete can effectively improve the load-carrying capacity and ductility of shaped columns [37]. Wang et al. [38] used highly ductile fiber concrete to reinforce the bottom of short cross-shaped columns in order to improve the seismic performance of short reinforced concrete crosses. It was found that the HDC reinforcement of short columns can effectively control the expansion of cracks, reduce the width of cracks, and improve the compression spalling phenomenon of concrete at the bottom of the columns, and the ductility and energy dissipation capacity of the concrete columns were significantly improved, the degradation of stiffness was slowed down, and the load-carrying capacity was also improved. This is because, although irregular cross-section columns are non-collinear, which may lead to an uneven stress distribution, HDC has a good toughness and crack resistance, which can adapt to this uneven stress distribution and reduce the phenomenon of stress concentration, thus improving the load-carrying capacity of the member. In addition, HDC has high corrosion and damage resistance, which means that HDC-reinforced irregular members are better able to resist erosion and damage from environmental factors, thus improving the durability of the structure.

In summary, domestic and foreign scholars have basically reached a consensus on the inclusion of fibers in HDC, and the inclusion of fibers such as PVA gives the concrete better ductility and integrity and better safety and durability compared to OPC. The single mixing of steel fibers, polyvinyl alcohol fibers, basalt fibers, and other fibers improves the flexural strength of the concrete, but reduces the compressive strength of the concrete, while the mixing of fibers can improve the situation to some extent. With the application of 3D printing technology in the field of construction, it promotes the development of HDC, which reduces the cost of materials, reflects the characteristics of green environmental protection, and conforms to the basic concept of sustainable development. In terms of practical application, HDC plays a crucial role in the reinforcement of components, especially for shaped columns.

#### 3.1.2. Mechanical Properties of HDC in Different Environments

Magnesium oxychloride cement (MOC) is an air-hardening cement made by mixing powdered magnesium oxide (MgO) with a concentrated solution of magnesium (Mg). Compared with ordinary silicate cement, MOC has many superior properties such as a high compressive strength, micro-expansion, early setting, good abrasion and corrosion resistance, and good bonding to different substances [39]. Wang et al. [40] analyzed the effect of fly ash and polyethylene fiber incorporation on the flowability of magnesium oxychloride cementitious composite material for cementitious engineering (MOC-ECC) and the tensile and compressive properties of MOC-ECC. It was shown that the tensile strength of MOC-ECC was more than 7 MPa, with obvious strain-hardening and multiple cracking characteristics.

Non-slump concrete (NSC) originated in the mid-18th century as a recommendation to use the least amount of water in concrete for optimal results [41]. The high conformal properties of NSC are widely used in crushed and precast concrete. Yuan et al. [42] investigated the synergistic effect of non-slump-free high-strength and high-ductility concrete (NSHSDC) with polyethylene (PE) and steel fibers (SF). Among them, the composites with a w/b of 16.8% were excluded due to their poor mechanical properties. It was found that the composites with a w/b of 17.2% showed a positive synergistic effect and had better mechanical properties.

The application of HDC is becoming more and more widespread because of its excellent properties. It has been found [43] that the fire temperature, the resting time, the cooling method, and the environment after cooling are important factors affecting the mechanical properties of concrete after high-temperature cooling. Deng et al. [44] designed 49 sets of cubic specimens and tested their compressive strength in order to study the compressive properties of HDC after high temperature, considering three factors: temperature, cooling and curing methods, and resting time. The results show that the temperature affects its compressive strength, which decreases with increasing temperature (see Figure 3). This is due to the fact that as the temperature increases, the water inside the HDC evaporates and the volume of the hydration products expands, resulting in an increase in the number of voids and cracks, which leads to a decrease in compressive strength. In addition, the increase in temperature causes the fibers inside the HDC to melt, leading to a decrease in compressive toughness. The cooling results are also different for different temperatures, with a negligible effect at a temperature of 200 °C. When the temperature is 400 °C, the resting time has a large effect on the cooling results (see Figure 4).

Extensive experimental and theoretical studies on the size effect of concrete have been conducted by scholars [45]. There are usually two main theories of scale effects, namely the law of statistical scale effects and the law of deterministic scale effects. Their representative theories were proposed by Weibull and Bazant, respectively, based on the fact that ultra-highly ductile concretes (UHDCs) have excellent ductility properties not found in normal concrete (NC). Ye et al. [46] explored the size effect of different specimen sizes, initial notches, and ductility levels on the bending behavior of UHDC. A number of three-point bending tests were performed on plain concrete beams and UHDC beams with different fiber volume fractions (1% and 2%) and different artificial notch sizes (notch/beam depth ratios of 0.10 and 0.23). It has been shown experimentally that under the effect of fiber bridging, the fracture toughness and the diameter of the machined zone of the material are increased by orders of magnitude, and then the size effect gradually diminishes and disappears during the transition of the material from brittle to tough. Finally, changes in the fracture processing zone (FPZ) of the material are utilized to account for the negligible size effect of the UHDC.

At present, scholars at home and abroad are focusing on the mechanical properties of HDC in different environments, mainly investigating its effects under different conditions of cement matrix, fiber doping, collapse, external environment and temperature, and cooling and curing methods. Conclusions have been obtained: (1) HDC damage is a typical ductile damage, the fibers are incorporated so that the concrete gets fracture toughness, and the bridging between the fibers means the concrete still shows good integrity even after damage; (2) Temperature affects the compressive strength of HDC, its compressive strength decreases continuously with the increase in temperature, and its color gradually changes to yellow-brown; (3) The size effect of HDC is diminished by the bridging of fibers.

### 3.2. Dynamic Mechanical Properties of HDC

Static mechanical properties, as the basis of the basic mechanical properties of concrete, have been studied by scholars for many years and significant results have been achieved. However, in the actual application of the project, the concrete working environment is often harsher and more complex. This involves not only the effects of static mechanical properties, but also the effects of external dynamic loads such as blast impacts, earthquakes, fires, and tsunamis. Therefore, the study of dynamic mechanical properties is of particular importance [47]. Impact loads can cause significant damage and present a high potential hazard. Destructive impact loads include trees, cars, or other objects thrown at building structures during hurricanes [48] and impacts on bridge piers by large vehicles [49]. Impact resistance is also one of the more pronounced of the dynamic mechanical properties. Therefore, it is very important to study the mitigation of damage from impact loads. Researchers [50] have conducted in-depth studies on various aspects of highly ductile concrete under impact blasting loads. Nicolaides [51] and others developed a new class of ultra-high-performance fiber-reinforced cementitious composites (UHP-CY). By mixing brass-coated steel fibers with diameters of 6 mm and 13 mm, the material has a high flexural strength, compressive strength, and energy-absorbing capacity. It was found that UHP-CY slabs significantly outperform high-performance concrete slabs under impact and blast loading. Ranade et al. [52] have found that high-strength high-ductility concrete (HSHDC) slabs maintain their load-carrying capacity and structural integrity under the impact of multiple drop hammers. Finite element analysis shows that even under the impact of the maximum energy in the study, the HSHDC plate only showed multiple fine cracks, fully demonstrating its excellent ductility. This is due to the inclusion of PE fibers in HSHDC and the fiber incorporation improving its ductility.

With the continuous development of concrete additives and mineral admixtures, the emergence of high-strength and high-performance concrete (HSHPC) has received more and more attention from scholars. Especially, it has been widely used in high-rise buildings, highway bridges, and component prefabrication [53]. The reason for this is that HSHPC offers more benefits compared to plain concrete, such as better compatibility, better erosion resistance [54], and being more in line with the concept of sustainable development. Scholars [55,56] have extensively studied the mesoscale material properties and uniaxial mechanical properties of HSHPC from a macroscopic perspective. Zhou et al. [57] investigated the mechanical properties of different types of high-strength high-performance concrete (HSHPC) with uniaxial compressive strengths of 47–90 MPa under uniaxial versus biaxial conditions. The addition of silica fume resulted in the formation of a denser microstructure within the HSHPC and improved the strength of the concrete. The uniaxial compressive strength of HSHPC is given as
(3)$$\left\{\begin{array}{l} a=0.968{\left(\frac{\left|f_{c}\right|}{100}\right)}^{2}-0.1835\left(\frac{\left|f_{c}\right|}{100}\right)+0.1206\\ b=a\\ c=-0.5847{\left(\frac{\left|f_{c}\right|}{100}\right)}^{2}-0.9819\left(\frac{\left|f_{c}\right|}{100}\right)-0.1867\\ d=-0.0386\frac{\left|f_{c}\right|}{100}+0.0927 \end{array}\right.$$

The biaxial compressive strength of HSHPC is
(4)$$\frac{\sigma_{1}}{f_{c}}=\left\{\begin{array}{l} c\cdot \left(\frac{\sigma_{1}}{\left|f_{c}\right|}\right)+d\left(\sigma_{1}/\sigma_{3}\geq -0.218\right)\\ a\cdot \left(\frac{\sigma_{1}}{\left|f_{c}\right|}\right)+b\left(\sigma_{1}/\sigma_{3}\geq -0.218\right) \end{array}\right.$$

σ_1_ and σ_3_ are the principal stresses in the biaxial direction. The tests showed that the uniaxial strength and stress ratios had a significant effect on the strength and deformation of HSHPC specimens subjected to biaxial compression and biaxial compression–tension loads, while they had little effect on specimens subjected to biaxial tensile loads.

Earthquakes are a typical event representative of external dynamic loading, and scholars have performed a lot of experimental research on the seismic performance of concrete in order to study the seismic-resistant concrete applied to buildings. Masonry structures strengthened by iron plate cladding [58] and fiber-reinforced polymer (FRP) [59] were subjected to vibration tests to evaluate their seismic performance. However, it is difficult to apply them on a large scale due to the increased cost and construction period. Deng et al. [60] analyzed the dynamic response and seismic performance of unreinforced masonry (URM) structures retrofitted with highly ductile fiber-reinforced concrete (HDC) overlays. High-strength and high-modulus PVA fiber with a 2% volume content was used. The vibration tests showed that the HDC overlay retrofit system improved the seismic performance of concrete more effectively than the welding wire reinforcement (WWR) overlay. Polymer and gelled composites typically exhibit brittle damage under tensile and flexural loading, which can be effectively addressed through the use of continuous reinforcement [61] or discrete reinforcement [62]. Farooq et al. [63] conducted tensile impact tests on high-toughness polymer composites (EDGCs). It was found that the 8 mm-long PVA fibers performed significantly better than the 13 mm-long ones, which may be due to the fact that the smaller size of the fibers allowed more fibers to be involved in the bridging interaction with the matrix, which improved its impact resistance. It was found that under dynamic loading, the thermally cured EDGCs exhibited ductile damage with dynamic strain coefficients ranging from 0.12 to 0.57 and good tensile properties.

The effect of strain rate on the strength and modulus of elasticity of plain concrete has been extensively studied [64]. As the strain rate increases, the growth rate of the strength of concrete in tension is about 2–3 times that in compression. The dynamic strength of high-strength concrete grows at a slower rate compared to normal concrete [65]. Scholars have carried out important research on the plausible causes of rate effects in concrete at different length scales. Ranade et al. [66] found that researchers at the University of Michigan have recently developed a high-strength and high-ductility concrete (HSHDC) with a good compressive strength and tensile ductility under quasi-static loads. The structural application of HSHDC is to cope with harsh extreme conditions. The structural application of HSHDC is designed to cope with severe extremes, and a micromechanics-based dimensional connection model [67] is shown in Equation (5) which analyzes the composite dimensional rate effect based on the microscopic dimensional rate effect. The fine-scale is based on the observed cracks through the aggregate rather than along the interface of the material at high strain rates [68], which effectively avoids misleading cracks generated by inertia at high strain rates.
(5)$$\sigma_{\left(\delta \right)}=\frac{V_{f}}{A_{f}}\int_{0}^{\pi /2}\int_{0}^{\left(L_{f}/2\right)\cos\left(\phi \right)}P\left(u\right)p\left(z\right)p\left(\phi \right)dzd\phi$$

Size correlation analysis shows that the rate effect of micro size adequately explains the rate-dependent effect on the composites. At the composite scale, the increase in matrix fracture toughness corresponds to the increase in first crack strength.

In summary, for scholars of the dynamic mechanical properties of the HDC drop hammer impact method, tensile impact test, and strain rate test, etc., the general conclusion is that HDC can still maintain the load-carrying capacity and integrity when subjected to the impact of loads many times. The HDC reinforcement of unreinforced structures, compared with welded wire reinforcement, shows a better seismic performance, OPC, the dynamic strength of the HDC grows faster, and the HDC is able to some extent to cope with extreme situations in practice.

## 4. Effect of Salt Erosion and Wet–Dry Cycling, Salt Freeze–Thaw Environments on HDC Durability

Concrete is most widely used in construction due to its excellent compressive strength and safety properties. Because northwestern China is mostly a sulfate-erosion and wet and dry cycle environment, the concrete will be subjected to different degrees of gradual damage to the internal structure of the concrete, resulting in the reduced durability of the building, and a serious bearing capacity decline will occur under major disasters. Therefore, it is important to study the effect on the durability performance of HDC under the action of dry and wet cycles versus salt freeze–thaw cycles.

### 4.1. Effect of Salt Erosion and Wet–Dry Cycling on the Performance of HDC

Highly ductile concrete (HDC) is a new type of fiber-reinforced concrete material developed on the basis of ECC (engineering cementitious composite). It follows the principles of fine mechanics and fracture mechanics, focuses on the bridge role of fiber and concrete, and has a good effect in structural seismic and reinforcement, which requires working in the harsh environment of HDC can meet the requirements. Salt erosion and wet–dry cycles are commonly encountered in harsh environments, so it is important to consider their impact on HDC. Current research on concrete in this environment focuses on mechanical properties and mechanism analysis. Previously, the main focus has been on the study of concrete in a single condition or continuously exposed to different environments [69,70]. Crack width is an important indicator of the erosion resistance of concrete during wet and dry cycling. Wu et al. [71] found that the increase in the chloride content of concrete during dry and wet cycling was not significant until the cracks in the concrete reached 90 μm, and the chloride content was a maximum when the cracks were greater than or equal to 90 μm and the dry-to-wet ratio was 1:3, and they proposed a chloride diffusion model with a predictive model for the remaining life. The final model predictions were found to be in good agreement with the experimental results, with the remaining life decreasing with increasing crack width.

Chloride erosion is one of the most serious problems affecting concrete structures, and in the case of cracked concrete when the width of the crack exceeds the threshold, it can cause a large diffusion of chlorides, destroying the internal structure of the concrete and making the structure less durable. Jang et al. [72] investigated the effect of crack width on the diffusion coefficient of chloride ions and found that the diffusion coefficient does not increase with the increase in crack width until the “threshold crack width” is reached. The threshold crack width is about 55–80 μm, and the diffusion coefficient only increases with crack width when the threshold width is exceeded. Therefore, crack control is one of the important factors affecting the durability of concrete, and highly ductile materials can solve this problem to a large extent. Strain-hardening cementitious composite (SHCC) is a high-ductility material with a good performance, and it was found that the penetration and corrosion rate of chloride in SHCC are much lower than that of ordinary concrete [73], which is due to the fact that SHCC, due to its good ductility, produces mostly fine cracks, which makes the chloride ions in the chloride salts diffuse slower, thus improving its durability. Paul et al. [74] investigated the performance of SHCC under chloride salt attack under dry and wet cycling conditions and compared it with plain concrete, which was determined by X-ray fluorescence spectrometry (XRF) analysis to determine the migration coefficients of chloride ions in SHCC [75], as shown in Equations (6) and (7).
(6)$$D_{cl}=\frac{RTh}{zFU}\frac{x_{d}-\alpha \sqrt{x_{d}}}{t}$$
(7)$$\alpha =2\sqrt{\frac{RTh}{zFU}}erf^{-1}\left(1-\frac{2c_{d}}{c_{0}}\right)$$

When 10% NaCl solution is used, (6) and (7) can be simplified to (8) and (9):(8)$$D_{cl}=\frac{\beta }{t}\left(x_{d}-2.53\sqrt{\beta x_{d}}\right)$$
(9)$$\beta =8.619\cdot 10^{-5}\frac{hT}{U}$$

Ultra-high-toughness cementitious composites (UHTCCs) are high-performance fiber-reinforced composites with pseudo-strain-hardening behavior (3% strain capacity) and a multiple cracking phenomenon (crack width 100 μm) [76]. UHTCC has excellent crack width control, resulting in a low water permeability and chloride permeability, which greatly improves the durability of concrete [77]. Zhao et al. [78] investigated the dynamic compressive properties of UHTCC under wet and dry conditions; the test found that the dynamic compressive strength of wet concrete is higher than that of dry concrete in the absence of fibers, and the strain rate sensitivity of UHTCC decreases when PVA fibers are incorporated, which is mainly due to the bridging effect of fibers in the matrix. Liu et al. [79] found that ECC can maintain its durability and high mechanical properties despite being in an eroded state under SO4^2−^ and Cl^−^, indicating that Cl^−^ has a limiting effect on sulfate erosion. Investigative studies have found that the bond strength between cement mortar and aggregate decreases significantly in some older buildings after years of salt–dry–wet cycling, which some researchers attribute to the physical or chemical effects of sulfate erosion [80]. This is due to changes in the internal structure of the HDC over time in a wet–dry cycling environment. Under physical deterioration, with the repeated infiltration and evaporation of water molecules, the pores and microcracks inside the HDC may further expand gradually, resulting in a decrease in the compactness and strength of the HDC, and the presence of water molecules may also promote the hydrolysis of substances in the HDC and the dislodgement of loose particles, which further exacerbate the process of physical deterioration. For example, during chemical erosion, when the sulfate solution enters the interior of the HDC, it will react chemically with the calcium hydroxide in the concrete, etc., to produce calcium alumina crystals, etc., and the volume of these compounds will expand, leading to stress concentration and damage inside the HDC. Luo et al. [81] tested the effect of sulfate attack on the mechanical properties of concrete by triaxial testing. The test results found that the specimens under low concentrations of sulfate treatment showed the characteristics of chemical sulfate erosion and the specimens under high concentrations of sulfate treatment showed the characteristics of physical sulfate erosion. This is mainly due to the relatively slow chemical reaction rate in low-sulfate-concentration environments, which produces a limited amount of expansion products such as caliche (AFt), gypsum, etc., and therefore behaves as CSA. The chemical reaction rate is accelerated in the high-sulfate-concentration environment, and the high concentration of sulfate rapidly crystallizes and precipitates in the concrete pores, generating a huge volume expansion, leading to cracks, and generating a large amount of anhydrous manganese at the cracks, and thus manifesting as PSA. To investigate the degradation characteristics of highly ductile concrete (HDC) under sulfate attack, Li et al. [82] used sulfate solutions with different concentrations for immersion and HDC specimens with different numbers of wet and dry cycles for uniaxial compressive strength tests. It was found that the compressive strength of HDC gradually increased up to 30 wet and dry cycles. After 105 wet and dry cycles, the compressive strength of HDC showed a linear decrease. The compressive strength of HDC decreases gradually with the increase in sulfate solution concentration (see Figure 5). The reason for this is that it is the result of a combination of both chemical and physical mechanisms. The chemical mechanisms are mainly twofold. First is the sulfate erosion reaction: sulfate ions (SO^2−^) will react with the hydration products (Ca(OH)_2_, C-S-H, etc.) in the HDC to produce expansion products such as Aft and gypsum, which leads to internal stresses within the HDC, resulting in a reduction of its strength, and with the increase in solution concentration, the faster the reaction rate, the more expansion products are generated, and the more the strength of the HDC is damaged, and the more serious the damage to the strength of the HDC. Then, the reaction of sulfate with calcium hydroxide in HDC consumes a large amount of OH^−^, which leads to a decrease in the alkalinity of the concrete and affects the stability of the components in the concrete, thus reducing the strength of the HDC. The physical mechanism is also divided into two aspects, one of which lies in the fact that sulfate is more likely to supersaturate in HDC and crystallize and precipitate when its concentration gradually increases. Volume expansion of the crystallization products puts pressure on the HDC pore walls, resulting in reduced strength. In addition, high concentrations of sulfate solutions are more permeable and can more easily penetrate into the interior of the HDC and create reactions that can lead to deterioration in the performance of the HDC. In addition, HDC’s resistance to both dry and wet cycling and sulfate attack is superior to that of ordinary concrete.

Du et al. [83] found that Cl^−^ does not completely control sulfate erosion, but immersing concrete in chloride and sulfate solutions can reduce the effects of sulfate erosion. Alyami et al. [84] found that a 10% sodium sulfate solution increased the erosion damage of concrete more significantly as compared to 30% and 5% sodium sulfate solutions in concentration. Yuan et al. [85], based on X-ray and CT theories, pointed out that the damage of concrete is exacerbated by dry and wet cycles and sulfate attack, and with the increase in the concentration of sulfate solution will further increase the degree of damage to the concrete, thus reducing its service life.

In summary, (1) HDC can control the generation and development of cracks to a large extent, and the cracks generated under chloride salt erosion are mostly fine cracks, which can slow down the diffusion rate of chloride ions; (2) Cl^−^ reduces the erosive effect of sulfate on concrete; (3) With the increase in the number of wet and dry cycles in the sulfate environment, the compressive strength of HDC shows an overall trend of increasing and then decreasing, and its resistance to sulfate erosion with wet and dry cycles is significantly better than that of plain concrete; (4) Increasing the concentration of sulfate solution gradually increases the rate of the loss of compressive strength of HDC, and its compressive strength is gradually reduced, which gradually reduces the durability and bearing capacity of highly ductile concrete under dry and wet cycles, and sulfate erosion is the joint result of physical and chemical effects on HDC.

### 4.2. Effect of Coupled Salt Freeze–Thaw Cycles on HDC Performance

In seasonally iced areas, concrete structures are usually subjected to freeze–thaw cycles and erosion by chloride and sulfate salts, posing durability challenges, which usually result in cracking and scaling [86]. Damage under freeze–thaw cycles leads to the intrusion of harmful ions such as chloride ions and sulfate ions, causing erosion within the structure, which has become one of the most important factors plaguing the service life of concrete. Yun et al. [87] investigated the effect of freeze–thaw on the tensile properties of HDC mixed with a 1.5% volumetric doping rate of PVA and PE fibers under single load, using dumbbell-shaped specimens (see Figure 6). The test results showed that the tensile strength was not affected much after 300 freeze–thaw cycles, but the high w/b tensile strain was affected more and showed a decreasing trend. This is due to the weak interfacial bonding of the fibers with the matrix at a high w/b and more pores; during freeze–thaw cycling, the water in the pores is more likely to freeze and produce volume expansion, leading to pore enlargement, connectivity, and the creation of new cracks, which reduces the tensile strain.

Currently, some researchers have investigated the effect of the roughness of the concrete matrix on the fracture properties [88] and fatigue properties [89] of concrete interfaces, and it was found that the rougher the interface the better the fracture properties and fatigue resistance of the interface. Tian et al. [90] investigated the shear strength degradation of ECC with different strengths at the interface with concrete under salt freeze–thaw cycling conditions and proposed an interfacial damage model for concrete, see Equation (10).
(10)$$D\left(N\right)=1-{\left[1-{\left(\frac{N}{N_{d}}\right)}^{1-c}\right]}^{\frac{1}{d+1}}$$

The test results show that the salt freeze–thaw cycle has a negative effect on the interfacial bond between ECC and concrete. In the salt freeze–thaw cycle environment, the lower strength of ECC and the smoother interface cause the earlier occurrence of the interfacial fracture phenomenon and will lead to the interface of the shear strength entering a sharp decline; the smoother the interface, the worse its resistance to erosion.

Moisture inside the concrete in the cold external environment can freeze and frost, which will lead to a reduction in the protective layer, causing erosion of the steel inside the concrete and reducing the service life of the concrete. The use of de-icing salt can reduce the freezing point of water, but the destruction of concrete under salt–freezing conditions will lead to the intrusion of chlorides, sulfates, etc. [91,92], which will greatly reduce the service life of the building, so the study of the effect of salt freeze–thaw cycles on the durability of concrete is of far-reaching significance. Li et al. [93] investigated the rate of the loss of mass and compressive strength of concrete with different basalt fiber admixtures under a chloride salt freeze–thaw environment. The test results show that with the increase in the basalt fiber (BF) admixture, its durability is stronger, and BF itself has an excellent corrosion resistance, chemical stability, and low-temperature resistance properties, so that the BF admixture improves the performance of concrete in these aspects. After 800 rapid salt freeze–thaw cycles, the amount of mass loss and compressive strength loss of concrete increased by about 1% and 2%, respectively. Scholars have investigated UHPC with different steel fiber dosages and different water-to-cement ratios in freeze–thaw cycling tests, and the results showed that the mass loss rate of UHPC showed a decreasing trend after a certain number of freeze–thaw cycles [94]. Lu et al. [95] investigated the effect of concrete composition on the frost resistance of UHPC. The test results showed that the freezing resistance of UHPC was better as the water–cement ratio decreased, which was attributed to the fact that the decrease in the water–cement ratio decreased the porosity inside the concrete, thus improving its freezing resistance. The freezing resistance of concrete changes with the change in UHPC components, although steel fiber and a mineral admixture can improve the freezing resistance of UHPC when they are mixed in appropriate amounts, but the biggest influence on it is the water–cement ratio. The mass loss rates were all less than 0.7% after 300 cycles. Gao [96] demonstrated that the relative dynamic modulus of UHPC was 100% after 800 cycles of freeze–thaw cycling with a water–cement ratio of 0.17, a steel fiber dosage of 7%, and steam curing at 90 °C. Lee et al. [97] chose the freeze–thaw cycle accelerated deterioration test for evaluating the durability performance of concrete and found that the relative dynamic modulus of concrete was still higher than 90% after 1000 freeze–thaw cycles when the water–cement ratio was 0.14 and the steel fiber admixture was 3%. Liu et al. [98] investigated the damage of crack-containing concrete in a salt–freezing environment, where the crack width of the concrete gradually increased, the mass loss increased, and the relative kinetic modulus gradually decreased with the number of salt freeze–thaw cycles. The relative dynamic modulus was calculated according to Equation (11).
(11)$$\tau_{n}=\frac{t_{0}-t_{c}}{t_{n}-t_{c}},E_{d}=\tau_{n}^{2}\times 100$$

For steel-fiber concrete, cracking occurs as soon as the stress exceeds the first cracking stress, which is closely related to the fracture of the aggregate, matrix, and ITZ [99]. The presence of cracks increases the chloride ion content and the depth of moisture influence within concrete exposed to sodium chloride solutions [100], which implies an increase in the detrimental effects on the damage resistance of concrete during salt freeze–thaw cycles.

From the above, it can be seen that (1) with the increasingly harsh environment of HDC service, purely considering the freezing and thawing under pure water can no longer be an effective study of its durability; therefore, in the long run, the study of its durability under the action of salt–freezing coupling is of greater significance; (2) The effect of HDC on the tensile strain is greater than the tensile strength under the coupling of salt freeze–thaw cycles showing a significant decreasing trend; (3) The rougher the interface of the concrete interface, the better the fracture properties and fatigue resistance of the interface; (4) The use of de-icing salt can reduce the freezing point of water, but the rate of loss of quality and compressive strength of the concrete under salt–freezing conditions increases, and the incorporation of BF can improve this situation; (5) In a certain range of the lower the water–cement ratio, the better the frost resistance of concrete, the reason being that the reduction of the water–cement ratio reduces the structure of the internal porosity which can significantly improve its durability; (6) Under the action of salt freezing and thawing cycles, the crack width and mass loss of concrete gradually increase, and the relative dynamic modulus of elasticity gradually decreases, while the incorporation of fibers can slow down the degree of damage.

## 5. Microscopic Studies and Engineering Applications of HDC

Scholars have conducted extensive research not only on the macro aspects of HDC, but also on the micro aspects. Because HDC has a higher ductility and crack resistance, it has great advantages in bending, cracking, earthquake resistance, and durability, so HDC will be more widely used in future projects.

### 5.1. Microscopic Studies

With the continuous progress of technology, the study of the properties of concrete from a macroscopic point of view can no longer adequately analyze its structural composition. The current methods for the microscopic study of concrete mainly include scanning electron microscopy (SEM), X-ray diffraction (XRD), infrared spectroscopy, and energy spectrum analysis techniques (EDS). Scholars [101] investigated the microstructure of concrete with ITZ by SEM and XRD and found that polypropylene fibers (PPFs) do not participate in any chemical reaction, and PPFs can significantly improve the microstructure of concrete by reducing the permeability and capillary porosity of concrete through a pore-filling effect, thus improving the durability of the concrete structure [102,103]. Yuan et al. [104] investigated the effect of glass fibers (GFs) and polypropylene fibers (PPFs) on the microstructural properties of concrete in relation to the water–cement ratio and fiber content. The SEM scale size is 20 μm. As can be seen from Figure 7, when the water–cement ratio (w/b) is 0.30 and 0.35, the aggregate and matrix are better bonded, and the cement matrix is very dense. In addition, the flowchart also showed obvious cracks, and the cracks appeared larger at 0.35 w/b. The reason is that when w/b becomes larger, the content of cementitious materials in the concrete decreases, which leads to a reduction in the strength and causes the appearance of larger cracks.

Figure 8 shows a schematic diagram of a specimen incorporated with GF (the w/b value of 0.30), which was summarized from a SEM with a scale size of 20 μm. Since GF is a mineral fiber material with good hydrophilicity, the bond between GF and the cement matrix is very tight [105]. The good fiber–matrix bond provides favorable conditions for GF to increase the concrete strength and reduce the water absorption of concrete. It can be noticed in the schematic diagram that hexagonal crystal products, acicular, prismatic crystals, and reticulated crystal structures can be clearly observed in the cement matrix. The GF surface is covered by a dense, hardened cement matrix, which acts as a pore filler, reduces porosity, and provides a significant improvement in bending and splitting tensile strength.

Figure 9 shows the schematic of the PPF-doped specimen at a w/b of 0.30, which is summarized from the SEM generalization with scale sizes of 20 μm and 400 μm, respectively. From the figure, it can be seen that PPF is loosely bonded to the matrix and reduces the strength of the concrete. Compared with GF, PPF as a synthetic fiber has hydrophobicity and the bond between PPF and the cement matrix is much lower than that of GF [105]. As can be seen in Figure 9, when the PPF admixture is 1.35%, PPF agglomeration phenomenon occurs, which reduces the denseness of the internal structure of the material and increases the porosity of the material, which in turn reduces the strength of the specimen and increases the water absorption of the concrete. Through the observation and analysis of SEM, w/b significantly affects the binding effect of GF and PPF; when w/b is 0.30, the interaction between the two is stabilized, and the effect is relatively poor when w/b is 0.35. Therefore, w/b should be reduced appropriately.

Ultra-high-strength ductile cementitious composites (UHS-UHDCCs) are building materials with a compressive strength greater than 160 MPa and pseudo-strain-hardening (PSH) uniaxial tensile ductility greater than 6%. PSH is the process by which cracking occurs in the form of more and finer cracks as the tensile stress increases [106]. Lei et al. [107] investigated an important link between the micro and macro values of UHS-UHDCC for compressive strength (>160 MPa) and uniaxial tensile ductility (>14.6%). When w/b was 0.23, the hydration products attached to the draw fibers showed a gradual increase as the amount of silica fume increased. When the dosage of silica fume was 40%, the hydration products on the fibers first increased and then decreased with the decrease in w/b. The most attached hydration products were found when w/b was 0.21, and as w/b kept decreasing, the more serious the fiber surface damage was, the fewer hydration products attached to the fibers. This is because the increase in the silica fume dosage and the decrease in w/b lead to an increase in the concrete compactness, strengthening the ITZ between the fibers and the mortar matrix, resulting in an increase in the interfacial bond strength.

The grooves left after the fibers were pulled out and their morphological changes varied. The differences in groove morphology were not significant when the silica fume dosage was varied from 0 to 30%. When the silica fume dosage increases and w/b decreases, the concrete compactness increases, which enhances the interfacial friction stress in the fiber pull-out process, making the fiber pull-out more difficult, resulting in the grooves changing from inconspicuous to gradually obvious.

It further improves the mechanical properties and durability of high-performance concrete, especially the shrinkage cracking resistance, which was tested and analyzed by Zhang et al. [108]. They investigated the interface between steel fibers and hydration products at different scanning magnifications with scale sizes of 5 μm and 10 μm from a microscopic point of view. An analysis of their micro studies summarizes the schematic shown in Figure 10. Some petal-like hydrated calcium silicate gel and some lamellar calcium hydroxide crystals can be seen on the fiber surface. These substances enhance the bond between the steel fibers and the cement paste and improve the mechanical properties of concrete, especially the flexural strength. The cracks were found to occur within the cementitious material or between the aggregate and the cementitious material. Further studies have revealed that the areas where the cracks appeared were C-H and ITZ, which also proved that C-H and ITZ are the weak areas of the concrete. The incorporation of steel fibers can effectively reduce the size of the bond layer and reduce the deterioration effect it produces.

Current research in the field of civil engineering focuses on the search for sustainable and economical materials; ultra-high-performance concrete (UHPC) is a new type of cementitious material with extremely excellent mechanical properties and durability [109,110], it is considered to be a sustainable and economical material for a wide variety of structures, and it is suitable for use in structures such as hydraulic structures, thin-layer structures, and marine structures [111]. Gu et al. [112], in order to study the damage of ultra-high-performance concrete (UHPC) under freeze–thaw action, examined the microstructure of a UHPC matrix before and after a freeze–thaw test by using the method of Mercury Impressed Porosity (MIP), scanning electron microscopy (SEM), and X-Ray Computed Tomography (X-Ray CT). Figure 11 shows the bond between the aggregate and cementitious material in ordinary UHPC, and Figure 12 shows the bond between the fiber and matrix in UHPC with 2% steel fiber incorporated. The bond strength between the mortar and the fiber matrix was observed to be excellent. Previous indentation studies have demonstrated that the micromechanical properties of UHPC are similar to those of the mortar [113], which suggests that no weak ITZs are present in UHPC. This is one of the reasons for the extremely good mechanical properties and erosion resistance of UHPC.

In summary, microanalysis has an important role in the field of concrete, mainly in the following aspects: (1) The microanalysis of HDC can help us to understand its properties such as strength, durability, and deformation characteristics more deeply. By looking at the aggregates in the concrete, the bonding of the fibers to the matrix, the number and width of cracks, the effect of mixing silica fume and other admixtures on the hydration products produced by the HDC, etc., we can more accurately assess the macroscopic aspects of its performance on this basis; (2) The results of micro-analysis can provide a scientific basis for the design of concrete materials and structures, and they can also be used to assess the quality of concrete and whether the construction process is reasonable and take timely measures to adjust and improve; (3) By observing the microstructural changes in concrete, we can determine the degree of concrete damage and formulate reasonable maintenance and repair programs accordingly; (4) Microanalysis can reveal the characteristics and behavioral laws of HDC at the microscopic scale; (5) The development and innovation of HDC can be continuously promoted through the study of concrete microstructure and properties.

### 5.2. Engineering Applications

Li et al. [114] introduced the mechanical properties and durability of ultra-high-toughness cementitious composites (UHTCCs), which have special mechanical properties, good durability, and versatility in the preparation process, so UHTCC has a better application prospect. Lightweight filler (LWF) is a material used in civil engineering and construction with a low deadweight and good workability. Zhang et al. [115] designed ECC mixes with a weaker matrix to allow for more cracking while maintaining a high strength, ductility, and light weight. Adding an air-entraining agent (AEA) and lightweight filler can effectively reduce the toughness and density of the matrix; the reason is that AEA will produce a large number of micro-bubbles after joining the reaction, which can effectively improve the working performance of the matrix, and in addition, LWF can effectively reduce the self-weight of the component, thus reducing the deformation of the matrix, improving the strain capacity and the saturated degree of cracking. Huang et al. [116] conducted a systematic study of a sprayable fiber-reinforced cementitious material with high ductility from material design to practical application. The material is molded by the wet mix spray coating process, and the mechanical properties of sprayable UHP concrete are higher than those of poured materials of the same proportion. The reason is that sprayable UHPC can form a dense concrete protective layer on the surface of the structure through the wet mix spray-coating process, which can effectively prevent the penetration of moisture, gas, and harmful substances and improve the compactness and strength of concrete. Qin et al. [117] investigated the flexural performance of reinforced concrete beams reinforced with high-strength and high-ductility engineering cementitious composite (HSHD-ECC). The test results show that HSHD-ECC can effectively improve the flexural performance of reinforced concrete beams. The reason is that the ECC layer has good synergistic properties with the steel reinforcement in the tension zone and the ordinary concrete in the compression zone of the RC beam, thus enhancing its load-bearing capacity. High-strength steel wires and polymer mortar are commonly used for the structural reinforcement of reinforced concrete members because of their high strength and good fire resistance. However, the polymer mortar is brittle under tension. In addition, the use of polymer mortar as a reinforcing layer will lead to secondary corrosion of the reinforcement. Yuan et al. [118] studied and proposed a method of flexural reinforcement of the existing reinforced concrete members using high-strength steel wires and high-ductility engineering cementitious composites (ECCs). The results showed that the flexural performance of the reinforced beams was significantly higher than that of the unreinforced control beams. The matrix type of the reinforcement had little effect on the damage mode of the beams, but had a greater effect on the cracking pattern of the beams. The reason for this is that high-tensile steel wire has a high strength and stiffness and ECC has a high ductility, high strength, and good durability; thus, the reinforcement of the components with high-tensile steel wire and ECC can significantly improve the bending capacity of the structure. Yu et al. [119] tested the mechanical properties of ultra-high-toughness cementitious composites (UHDCCs) at three levels: material, member, and structural, in order to verify the feasibility of their use in steel-free reinforced buildings. The test results revealed that the flexural and shear deformation resistance of UHDCC beams was superior to that of the reference reinforced concrete (RC) beams. The performance of UHDCC frames was found to be able to meet the requirements of various seismic codes. The feasibility of steel-free reinforced buildings was preliminarily confirmed.

Fiber mesh high-ductility concrete is a fiber-reinforced composite material based on the design principle of micromechanics with cement, quartz sand, etc., as a matrix. Song et al. [120,121] investigated the flexural performance of a fiber mesh with fabrics—high-ductility concrete (TR-HDC)-reinforced precast hollow core slabs with reinforced concrete slabs. The tests showed that the use of TR-HDC significantly increased the flexural load capacity, cracking, yield strength, and peak load of the slabs. One of the reinforced prefabricated hollow core slabs showed an increase in the number of cracks, but the crack width was reduced and the cracks appeared to be numerous and dense. The reason for this is that with the load in the span near the middle of the first crack, the crack cross-section of the concrete has to bear the tensile force transmitted to the longitudinal bars and fiber mesh, and a small amount of force borne by the fibers in the bridging role of the cracks will continue to appear, but because of the role of the fibers in the bridging role of the cracks these will appear to be fine with dense characteristics.

The reclamation and recycling of materials is an important method of sustainable development, which helps to realize the recycling of resources and the sustainable development of the environment. This makes scholars consider this aspect of research when studying how to improve the performance of concrete. Song et al. [122] investigated the mechanical properties of fiber-reinforced recycled micronized concrete (FRPC) and optimized the mix ratio of FRPC through orthogonal tests. The results show that the ductility of FRPC is the best when the substitution rate of recycled micronized powder is 45%, and the mechanical properties of FRPC reach the best values when the volume doping of basalt fiber is 0.1%. This is due to the high friction coefficient of BF, and its uniform distribution in FRPC can form a good fiber–cement matrix lattice structure, which plays the role of toughening the concrete, but if too much BF is doped, it will reduce the working performance of FRPC, increase the porosity, and reduce the compressive strength. The alkaline content of RCP is high, and the C-S-H cementitious material produced by the reaction can fill the internal pores of the matrix and improve the microporous structure of concrete. The strain of FRPC increases with the increase in the RCP substitution rate, and it reaches the maximum value when the substitution rate of RCP is 45%. When the substitution rate is greater than 45%, it results in a significant reduction in the amount of cement, leading to a lower C-S-H cementitious content of the cement hydration products, making the tensile strength lower. For the new building materials, some scholars have proposed unreinforced ultra-high-ductility concrete, which was mechanically tested at three levels: material, component, and structure, and the test results showed that it has a good durability and good flexural properties. Zhang Luo et al. [123] analyzed the design and construction of the exhibition hall structure of the Bamboo and Rattan Pavilion of the 10th China Flora Expo in 2021, which is the first time in China that the unreinforced ultra-high-ductility concrete (UHDC) material has been used completely in a building structure and constructed without reinforcement. The shell stress under the design load is less than the design strength of the material, which can meet the basic requirements of unreinforced design, simplify the construction work, and reduce the production cost.

In conclusion, HDC as a class of high-performance concrete exhibits an excellent durability performance which gives it a wide range of application prospects, and to a certain extent it can replace the application of steel reinforcement. To alleviate the pressure on the application of steel in the project, the current research for the further practical application of HDC provides a method, but it also provides a new route for the implementation of the concept of sustainable development.

## 6. Conclusions

The research focuses on a cement-based composite high-ductility concrete made of gelling materials, mineral admixtures, water, fibers, and other admixtures. The key characteristics of HDC, such as its mechanical properties, durability, and microstructure, are considered, the influence of different conditions on HDC performance is summarized, and its advantages and disadvantages are analyzed. The following is a comprehensive study and analysis of the results of the study.

At present, HDC has not been fully utilized, its quality cannot be guaranteed in actual projects, the construction is difficult, and it is difficult to accurately use in different construction processes and different environments. This is still an important problem facing its use; in response to the above problems, researchers can use the factory production method to carry out a special production line to ensure its production quality.Ordinary HDC has a tensile strain capacity (strain at peak tensile stress) of 3–5% and a tensile strength of 3–7 MPa. Although HDC has a good tensile strain capacity, its ductility is not sufficient to ensure that it can be used as a structural material without reinforcement, especially under extremely harsh conditions.The mechanical anisotropy and interlayer bond strength of 3D-printed UHDC need to be studied more systematically from both experimental and numerical aspects. In addition, it is also necessary to consider the introduction of bionics in 3D-printed structures, and the design of different types of release structural components, such as bamboo columns, pearl beams, honeycomb panels, etc. For the 3D printing of special-shaped components, the shotcrete process is used, which makes the structure formation simpler and faster, eliminating the process of removing the external template.At present, there is insufficient research on the effect of the mechanical properties of HDC after high temperature in terms of the resting time, and the strength degradation law of fiber-reinforced cementitious materials under the mutual influence of factors such as temperature, cooling mode and resting time has not been adequately investigated. Also, there is still a need to deepen the research on the consideration of HDC’s resistance to dry and wet cycling and sulfate erosion.Research on the HDC of salt erosion under the coupling of wet and dry cycles and freeze–thaw cycles can help engineers to more accurately assess the safety of the structure, identify potential safety hazards in a timely manner, and provide targeted maintenance measures for the existing concrete structure, thus improving the quality and service life of the concrete structure and reducing the high repair and reconstruction costs at a later stage.The fiber-mesh-reinforced cement mortar (TRM) reinforcement technique can improve the stress performance of members, but there are certain defects, such as low matrix elongation, a large crack width, and no longer transferring loads after cracks appear.

## Figures and Tables

**Figure 1 materials-17-04596-f001:**
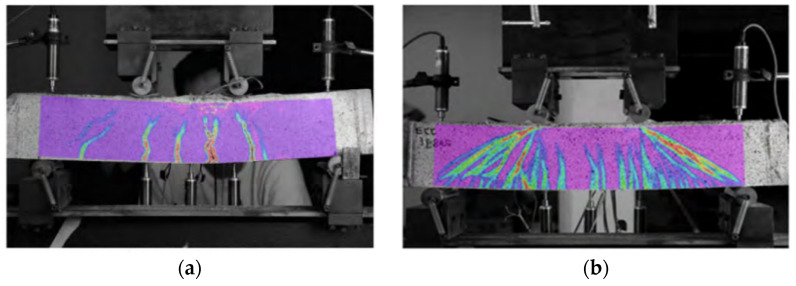
Schematic diagram of beam cracking in a four-point bending test. (**a**) Schematic diagram of plain concrete [19]. (**b**) Schematic diagram of UHDC [19].

**Figure 2 materials-17-04596-f002:**
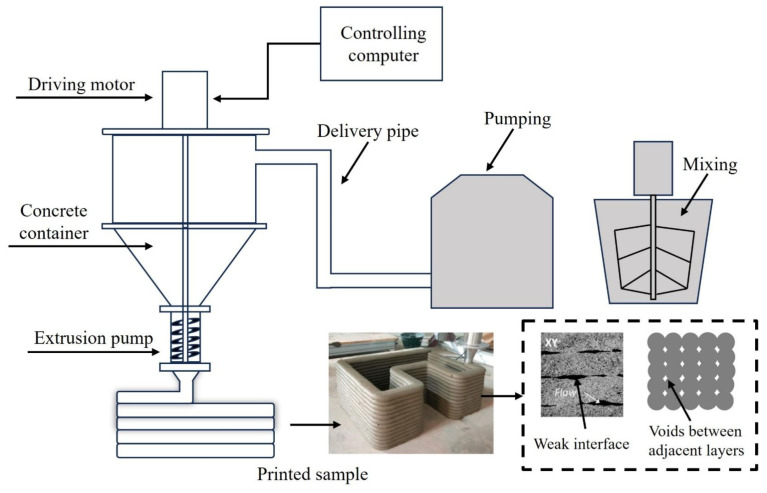
Whole process of 3D concrete printing [33].

**Figure 3 materials-17-04596-f003:**
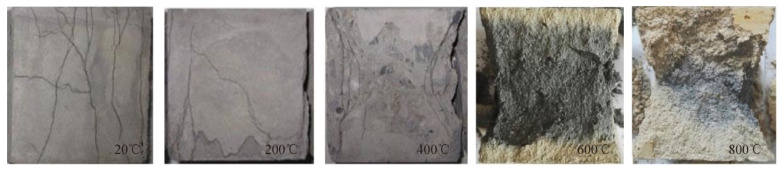
Failure modes of HDC cubes exposed to elevated temperatures after compression [44].

**Figure 4 materials-17-04596-f004:**
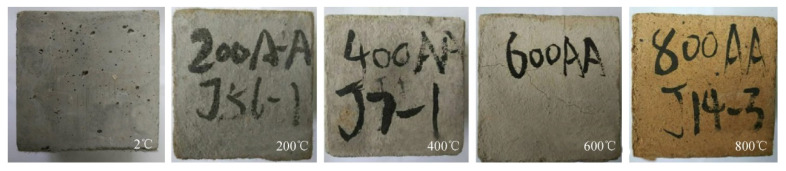
Color change of HDC cubes exposed to temperatures [44].

**Figure 5 materials-17-04596-f005:**
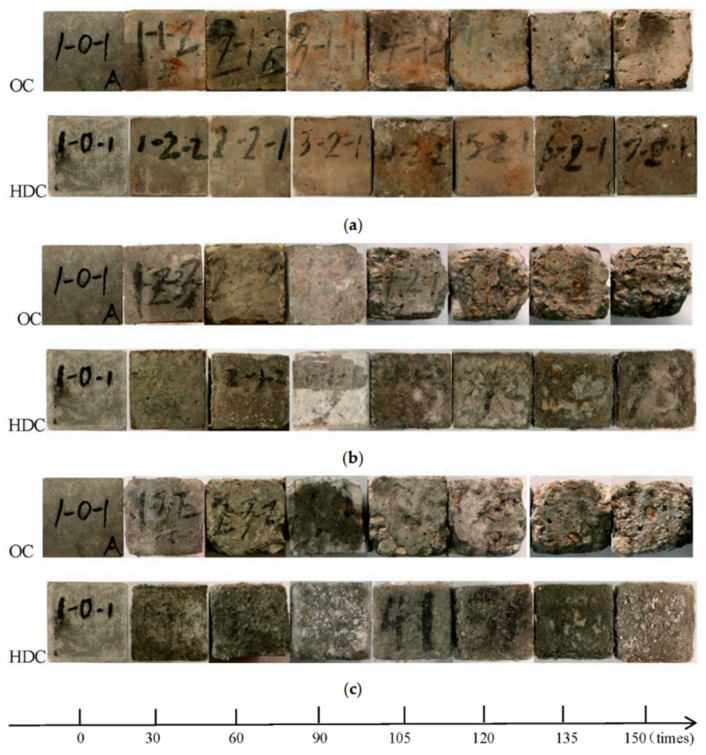
Morphology of HDC and OC samples with different numbers of wet–dry cycles in sulfate solution: (**a**) 5% sulfate solution; (**b**) 10% sulfate solution; (**c**) saturated sulfate solution [82].

**Figure 6 materials-17-04596-f006:**
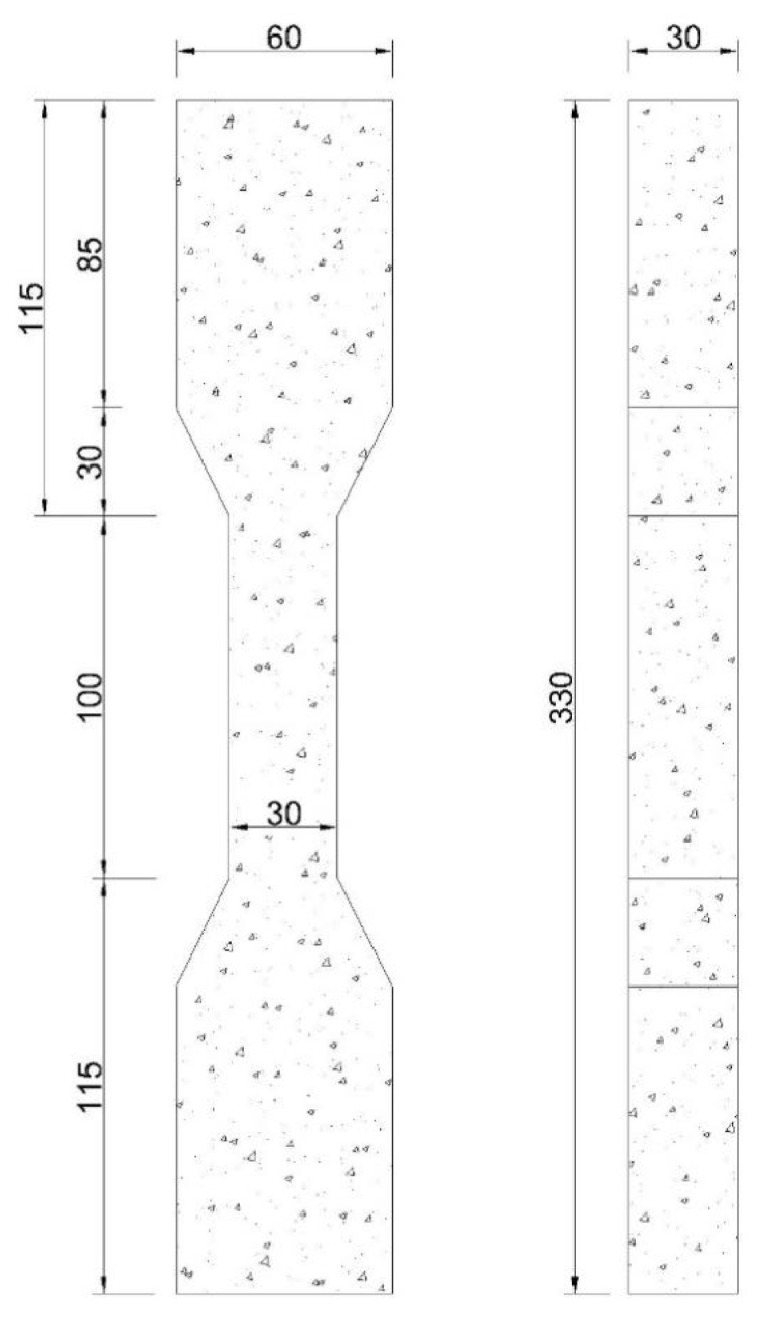
Dimensions of a dumbbell-shaped tensile specimen. Dimensions (unit: mm).

**Figure 7 materials-17-04596-f007:**
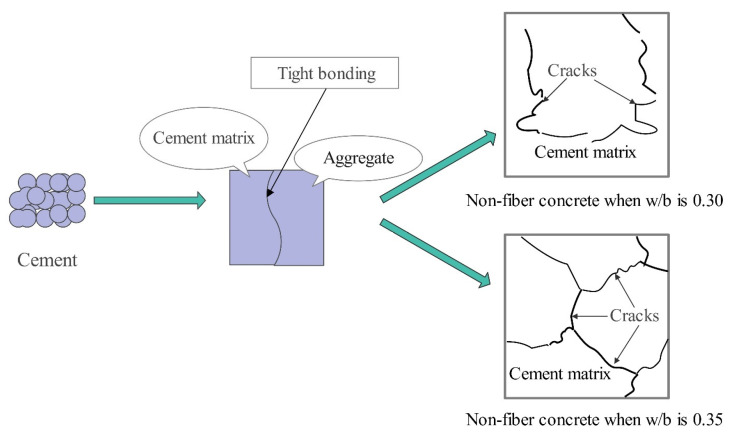
Schematic diagram of microanalysis of non-fiber concrete.

**Figure 8 materials-17-04596-f008:**
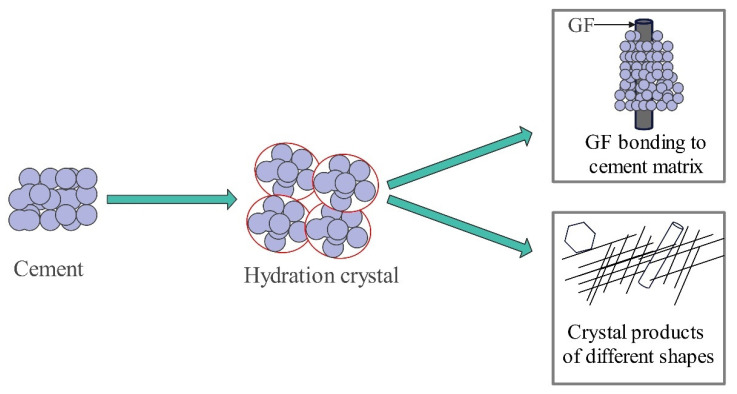
Microscopic schematic of GFRC at W/b of 0.30.

**Figure 9 materials-17-04596-f009:**
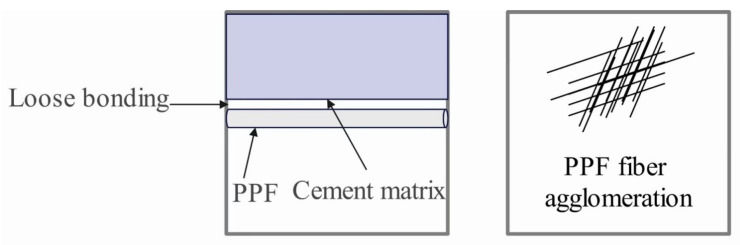
Micro-schematic of GFRC at W/b of 0.30.

**Figure 10 materials-17-04596-f010:**
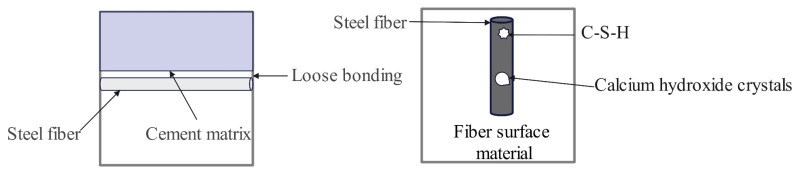
Schematic representation of the interface between steel fibers and aqueous products.

**Figure 11 materials-17-04596-f011:**
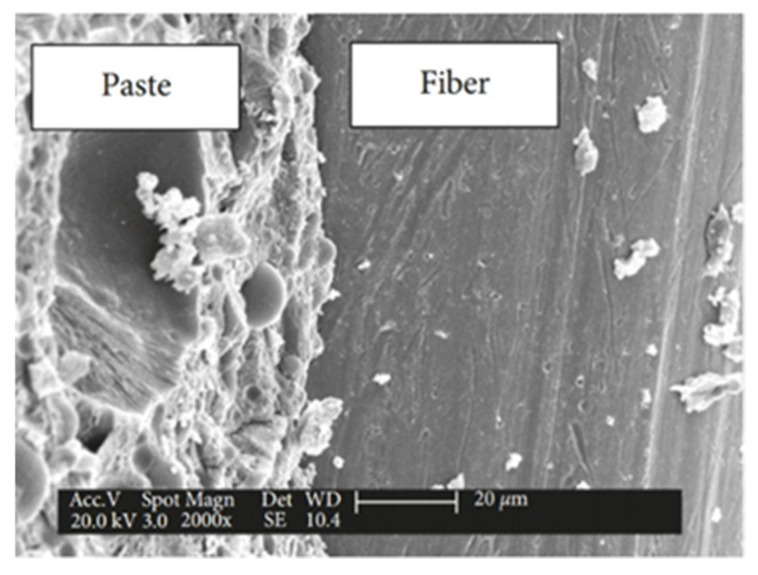
The bond between the sand and paste in plain UHPC [112].

**Figure 12 materials-17-04596-f012:**
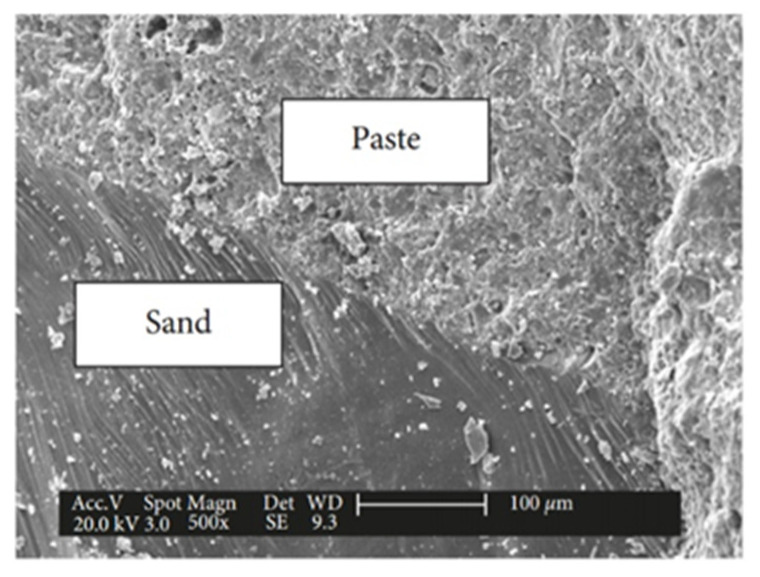
The bond between the fiber and the matrix in UHPC-2% [112].

**Table 1 materials-17-04596-t001:** Common fiber parameters [22].

Fiber Name	Length (mm)	Diameter (μm)	Density (g/cm^3^)	Tensile Strength (MPa)	Fusing Point (°C)	Modulus of Elasticity (Gpa)
PVA	3~15	12~39	1.32	1600~2500	255~230	40~80
PE	3~15	7~35	0.96	2500~2800	124~138	70~80

**Table 2 materials-17-04596-t002:** Strength test results and strength-effectiveness on HSFRC and HSC [23].

Fiber Volume Fraction (%)	Compressive Strength		Splitting Tensile Strength		Modulus of Rupture	
	Measured (MPa)	Strength-Effectiveness (%)	Measured (MPa)	Strength-Effectiveness (%)	Measured (MPa)	Strength-Effectiveness (%)
0	85	-	5.8	-	6.4	-
0.5	91	7.1	6.9	19.0	8.2	28.1
1.0	95	11.8	8.7	50.0	10.1	57.8
1.5	98	15.3	10.8	86.2	12.3	92.2
2.0	96	12.9	11.5	98.3	14.5	126.6

Strength-effectiveness $=\frac{HSFRC\;strength-HSC\;strength}{HSC\;strength}$ × 100%.

**Table 3 materials-17-04596-t003:** Comparison of predicted and measured values for compressive and splitting tensile strengths and modulus of rupture [23].

Fiber Volume Fraction (%)	Compressive Strength			Splitting Tensile Strength			Modulus of Rupture		
	Predicted (MPa)	Measured (MPa)	Prediction Error (%)	Predicted (MPa)	Measured (MPa)	Prediction Error (%)	Predicted (MPa)	Measured (MPa)	Prediction Error (%)
0	85	85	0	5.8	5.8	0	6	6.4	0
0.5	91	91	0	7.3	6.9	5.80	8.2	8.2	0
1.0	95	95	0	8.8	8.7	1.15	10.2	10.1	0.99
1.5	97	98	−1.02	10.3	10.8	−4.63	12.3	12.3	0
2.0	96	96	0	11.7	11.5	1.74	14.5	14.5	0

Prediction error $=\frac{predicted\;value-measured\;value}{measured\;value}$ × 100%.

## Data Availability

Data are contained within the article.

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
