# Peer review of "Advances in Highly Ductile Concrete Research"

_materials, 2024, doi:10.3390/ma17184596_

Round 1

Reviewer 1 Report

Comments and Suggestions for Authors

The advances in highly ductile concrete research are presented in this work. Before it is published, I suggest some improvements.

In the abstract more information about the results analysed should be added. Add in each section the main results that are presented in the work.
The variables presented in the equations and, mainly the acronyms should be added at the beginning or at the end of the paper.

In the equations, a space before and after should be added and the number of the equations should be placed on the left side. Place the number (1) in the line 167 instead of the number (0). In the equation in line 124, the same rules can be applied.

The figures should be improved. Increase their dimensions and quality, as an example. See Figures 2, 3, 4, 7, 9, 10, 12 and 13.

In the figures, as an example Figure 3, the whole process of 3D concrete printing, more explanation should be added in the text. In other figures, the scheme, as presented in Figure 3, can be added.

In the results presented (in text and figures), it will be interesting to present the input data used to obtain the results. Use a table, if there is a lot of information, or add it to the figure caption.

The conclusions should be shortened. Focus on the main conclusions.

Comments on the Quality of English Language

Minor editing of English language required.

Author Response

Comments 1: [In the abstract more information about the results analysed should be added. Add in each section the main results that are presented in the work.]

Response 1: Agree. We have completed the revisions to emphasize this point. We add more information about the results of the analysis. This change can be found in the revised manuscript – page 1, abstract, and lines 12-22.]

Comments 2: [The variables presented in the equations and, mainly the acronyms should be added at the beginning or at the end of the paper.]

Response 2: Agree. We have completed the revisions to emphasize this point. We added the variables given in the equations at the end of the paper. This change can be found in the revised manuscript – page 27, Nomenclature, and lines 910-930.]

Comments 3: [In the equations, a space before and after should be added and the number of the equations should be placed on the left side. Place the number (1) in the line 167 instead of the number (0). In the equation in line 124, the same rules can be applied.]

Response 3: Agree. We have completed the revisions to emphasize this point. We have put a space before and after, in the equation, and the numbering of the equation has been placed on the left side, changing the number (0) to (1). This change can be found in the revised manuscript – page 5, Paragraph 1, and line 181.]

Comments 4: [The figures should be improved. Increase their dimensions and quality, as an example. See Figures 2, 3, 4, 7, 9, 10, 12 and 13.]

Response 4: Agree. We have completed the revisions to emphasize this point. We have increased the size and quality of the images. This change can be found in the revised manuscript – page 5,6,8,9,12,15,17,17, Paragraph 2,1,1,1,1,1,1, 2, and line 186,206,266,304,398,519,572,574.]

Comments 5: [In the figures, as an example Figure 3, the whole process of 3D concrete printing, more explanation should be added in the text. In other figures, the scheme, as presented in Figure 3, can be added.]

Response 5: Agree. We have completed the revisions to emphasize this point. For the whole process of 3D concrete printing, we have added more explanations in the main text. This change can be found in the revised manuscript – page 6, Paragraph 1, 2, and lines 198-203.]

Comments 6: [In the results presented (in text and figures), it will be interesting to present the input data used to obtain the results. Use a table, if there is a lot of information, or add it to the figure caption.]

Response 6: [However, the article did not have much results and data, so tables were not used to present the results used to obtain them.]

Comments 7: [The conclusions should be shortened. Focus on the main conclusions.]

Response 7: Agree. We have completed the revisions to emphasize this point. We have shortened the conclusions to highlight the main findings. This change can be found in the revised manuscript – page 26, Paragraph 3, and lines 871-908.]

8. Response to Comments on the Quality of English Language

Point 1: We have made improvements to the quality of the English language.

Response 1: For more details, please check the main text.

Reviewer 2 Report

Comments and Suggestions for Authors

materials-3186220:

*Many phenomena are not explained in the paper. Review the paper by explaining the following points in detail:

1. The article mentions the importance of microscopic studies of HDC but does not explain in detail which specific aspects of these studies are essential to understanding the properties of concrete at the macroscopic level.

2. Although there is discussion about the effects of different types and amounts of fibers in concrete, there is a lack of explanation about the exact mechanisms by which these fibers influence the material's ductility and strength.

3. The article refers to the bonding between fibers and the cement matrix, especially with glass and polypropylene fibers, but does not detail the factors that affect the quality of this interaction or how it can be optimized.

4. The article mentions the impact of temperature on the compressive strength of concrete but does not explain the underlying mechanisms that cause the decrease in strength with increasing temperature.

5. The article discusses the importance of the dynamic properties of HDC, especially under impact loads, but does not provide details on how different HDC compositions may influence these dynamic outcomes.

6. The article mentions that HDC strength decreases with increasing sulfate solution concentration but does not explain the chemical or physical mechanisms responsible for this degradation.

7. While the article mentions the impact of different environmental conditions, such as freeze-thaw cycles, it does not explore how these specific conditions interact with the composite materials within HDC.

8. There is a brief mention of HDC behavior under shear loads, but the article does not provide a detailed explanation of how HDC responds to these types of stresses compared to traditional concrete.

9. The article discusses the effects of wetting and drying cycles on the durability of HDC but does not detail how these conditions affect the material's microstructure over time.

10. Although there is mention of the advantages of using HDC in columns with irregular cross-sections, there is a lack of explanation about how HDC specifically improves the load-bearing capacity and durability of these structures compared to traditional materials.

*To strengthen the findings of the article and ensure that the conclusions are robust and reliable, the following statistical analyses could be included or better detailed:

1. Analysis of Variance (ANOVA): To determine if there are statistically significant differences between the different experimental groups, especially when comparing the performance of HDC with different types or concentrations of fibers.

2. Confidence Intervals: To provide an estimate of the precision of the obtained results, it is important to include confidence intervals for key metrics such as strength.

3. t-Test for Independent Samples: To compare the means of two different samples, such as HDC with different types of fibers, to verify if the observed differences are statistically significant.

4. Sensitivity Analysis: To assess the robustness of the results and understand how small changes in experimental conditions or model assumptions might impact the outcomes.

Including these statistical analyses would help validate the results presented, increase confidence in the conclusions, and provide a more detailed understanding of the effects and interactions between the studied variables.

Comments on the Quality of English Language

Moderate editing of English language required.

Author Response

Comments 1: [The article mentions the importance of microscopic studies of HDC but does not explain in detail which specific aspects of these studies are essential to understanding the properties of concrete at the macroscopic level.]

Response 1: Agree. We have completed the revisions to emphasize this point. We explain in detail that by observing the aggregates in concrete, the bonding of fibers to the matrix, the number and width of cracks, the effect of the inclusion of admixtures such as silica fume on the production of HDC hydration products, etc. is essential to understand the properties of concrete at the macro level. This change can be found in the revised manuscript – page 24, Paragraph 1, and lines 763-768.]

Comments 2: [Although there is discussion about the effects of different types and amounts of fibers in concrete, there is a lack of explanation about the exact mechanisms by which these fibers influence the material's ductility and strength.]

Response 2: Agree. We have completed the revisions to emphasize this point. We have added the explanation of the exact mechanism by which fibers affect the ductility and strength of the material. This change can be found in the revised manuscript – page 2, Paragraph 2, and lines 75-82.]

Comments 3: [The article refers to the bonding between fibers and the cement matrix, especially with glass and polypropylene fibers, but does not detail the factors that affect the quality of this interaction or how it can be optimized.]

Response 3: Agree. We have completed the revisions to emphasize this point. We complement the bonding of glass and polypropylene fibers by detailing the factors that influence the quality of this interaction or how it can be optimized. This change can be found in the revised manuscript – page 21, Paragraph 1, and lines 682-685.]

Comments 4: [The article mentions the impact of temperature on the compressive strength of concrete but does not explain the underlying mechanisms that cause the decrease in strength with increasing temperature.]

Response 4: Agree. We have completed the revisions to emphasize this point. We add potential mechanisms leading to a decrease in intensity with increasing temperature. This change can be found in the revised manuscript – page 8, Paragraph 1, and lines 278-282.]

Comments 5: [The article discusses the importance of the dynamic properties of HDC, especially under impact loads, but does not provide details on how different HDC compositions may influence these dynamic outcomes.]

Response 5: Agree. We have completed the revisions to emphasize this point. We add details on how different HDC compositions affect these dynamic results under shock loading. This change can be found in the revised manuscript – page 10,11, Paragraph 1, 2, and lines 330-331,349-350,366-367,373-376.]

Comments 6: [The article mentions that HDC strength decreases with increasing sulfate solution concentration but does not explain the chemical or physical mechanisms responsible for this degradation.]

Response 6: Agree. We have completed the revisions to emphasize this point. HDC strength decreases with increasing sulfate solution concentration, and we explain the chemical or physical mechanisms that lead to this degradation. This change can be found in the revised manuscript – page 14, Paragraph 1, and lines 487-493,500-518.]

Comments 7: [While the article mentions the impact of different environmental conditions, such as freeze-thaw cycles, it does not explore how these specific conditions interact with the composite materials within HDC.]

Response 7: Agree. We have completed the revisions to emphasize this point. We complemented this by exploring how different environmental conditions (e.g., freeze-thaw cycles) interact with the composites within the HDC. This change can be found in the revised manuscript – page 16,17, Paragraph 1,1, and lines 549-555,584-589.]

Comments 8: [There is a brief mention of HDC behavior under shear loads, but the article does not provide a detailed explanation of how HDC responds to these types of stresses compared to traditional concrete.]

Response 8: Agree. We have completed the revisions to emphasize this point. We explain how HDC responds to shear and tensile stresses compared to conventional concrete. This change can be found in the revised manuscript – page 2, Paragraph 1, and lines 55-59.]

Comments 9: [The article discusses the effects of wetting and drying cycles on the durability of HDC but does not detail how these conditions affect the material's microstructure over time.]

Response 9: Agree. We have completed the revisions to emphasize this point. We illustrate how the microstructure of the material is affected over time under dry and wet cycling conditions. This change can be found in the revised manuscript – page 13, Paragraph 2, and lines 470-482.]

Comments 10: [Although there is mention of the advantages of using HDC in columns with irregular cross-sections, there is a lack of explanation about how HDC specifically improves the load-bearing capacity and durability of these structures compared to traditional materials.]

Response 10: Agree. We have completed the revisions to emphasize this point. We explain how HDC specifically improves the load-bearing capacity and durability of irregular-section columns compared to conventional materials. This change can be found in the revised manuscript – page 7, Paragraph 1, and lines 229-235.]

11. Response to Comments on the Quality of English Language

Point 1: We have made improvements to the quality of the English language.

Response 1: For more details, please check the main text.

Reviewer 3 Report

Comments and Suggestions for Authors

 Please write the effects that have the type, content, and geometry of fibers on the mechanical properties and durability of highly ductile concrete (HDC)?

Please write the behavior of HDC when exposed to more complex loading conditions that replicate real-world environments, including multiaxial stress states or fatigue loading?

Please provide in text the economic and environmental factors that should be evaluated when comparing HDC to standard concrete.

Write in text if it is feasible to include recycled or waste materials in HDC formulations without affecting their performance.

Please provide more details in text about Figure 1.

Please change the resolution of Figure 2.

Please change the resolution of Figure 4.

Please refer to the strategies that can be employed to enhance modeling and simulation tools for more precise predictions of HDC behavior and improved mix design optimization.

Author Response

Comments 1: [Please write the effects that have the type, content, and geometry of fibers on the mechanical properties and durability of highly ductile concrete (HDC)?]

Response 1: Thank you for pointing this out. We agree with this observation and our response is as follows. [Different kinds of fibers have different effects on the mechanical and durability properties of HDC. For example, steel fiber has high modulus and tensile properties, therefore, it can improve the compressive strength and tensile strength of HDC. Polypropylene fibers, polyvinyl alcohol fibers, etc., on the other hand, significantly improve the toughness and crack resistance of HDC through their good dispersibility and crack resistance. Although steel fibers have high strength, they are easy to rust and are not easy to mix uniformly, which is not conducive to the long-term performance of HDC. Organic fibers such as polyvinyl alcohol fibers have better alkali resistance and aging resistance, and can maintain the stability of HDC for a long time. Fiber dosage has a significant effect on the mechanical properties and durability of HDC, the appropriate amount of fibers can reduce the stress concentration within the concrete, and can give full play to the role of the bonding matrix, significantly improve the mechanical properties and durability of the concrete, but when too much fiber is added, it will lead to a decrease in the fluidity of the concrete and increase the porosity of the concrete, resulting in difficulties in the construction, and the durability will also be reduced. Fiber geometry (e.g., length, diameter, aspect ratio, etc.) has a significant impact on the mechanical and durability properties of HDC. Longer fibers usually provide better bridging, but may agglomerate during construction. Fibers with a larger L/D ratio are able to distribute loads more efficiently and reduce stress concentrations within the HDC, thereby increasing its durability. Fiber diameter also affects its distribution and performance in the HDC. Finer fibers are more likely to be evenly dispersed in the HDC, but may not have as good mechanical properties as coarser fibers.]

Comments 2: [Please write the behavior of HDC when exposed to more complex loading conditions that replicate real-world environments, including multiaxial stress states or fatigue loading?]

Response 2: Thank you for pointing this out. We agree with this observation and our response is as follows. [(1) Under multiaxial stress states, HDCs usually exhibit higher strength than under uniaxial stresses because the stresses in different directions may interact and reduce the tendency of HDCs to damage.

(2) HDC exhibits more obvious strain hardening phenomenon under multiaxial stress state, which indicates that the strain of HDC increases with the increase of stress, but there is no sudden brittle damage, resulting in better ductility and toughness of HDC when subjected to multiaxial stress.

(3) In general, HDC may exhibit a longer fatigue life than ordinary concrete under fatigue loading due to its high ductility and good strain capacity.

(4) During the fatigue loading process, the HDC will gradually accumulate internal damage, such as the expansion of microcracks and the fracture of fibers. The appearance of these damages will lead to a gradual decrease in the stiffness and strength of the HDC until its final destruction.

(5) HDC is mixed with reinforcing materials such as fibers, which can inhibit the emergence and expansion of fatigue cracks, thus improving the fatigue resistance of concrete. Therefore, under the same conditions, HDC has better fatigue resistance than ordinary concrete.]

Comments 3: [Please provide in text the economic and environmental factors that should be evaluated when comparing HDC to standard concrete.]

Response 3: Thank you for pointing this out. We agree with this observation and our response is as follows. [(1) As far as material cost is concerned, the raw materials of HDC may contain special types of cement, fiber materials, mineral admixtures, admixtures, etc., and the price of these materials may be higher than that of the materials used in ordinary concrete, and the production of HDC is more complicated than that of ordinary concrete, and the cost of production may be high, but the HDC has a higher durability and damage-resistant ability, which makes the maintenance cycle of HDC longer, and its post maintenance cost is much lower than that of ordinary concrete. maintenance cost is much lower than that of ordinary concrete. Taking into account the cost of materials, construction and maintenance, although the initial investment of HDC may be higher than that of ordinary concrete, the advantages are more significant in terms of long-term economic benefits.

(2) Due to its excellent performance, HDC can greatly reduce the loss of resources in the process of subsequent reinforcement and maintenance, and also reduce the loss of materials due to frequent maintenance and replacement, reduce the generation of waste, and effectively control carbon emissions, which has a greater advantage in terms of environmental protection and sustainability.]

Comments 4: [Write in text if it is feasible to include recycled or waste materials in HDC formulations without affecting their performance.]

Response 4: Thank you for pointing this out. We agree with this observation and our response is as follows. [It is feasible to incorporate recycled or waste materials without affecting the performance of HDC, but this requires detailed research and experimental verification. The following is a detailed analysis of this issue:

(1) First, recycled or discarded materials suitable for incorporation into the HDC need to be selected. These materials should be pre-treated to remove impurities, contaminants or hazardous substances from them to ensure that they do not negatively affect the performance of the HDC. For example, waste plastics and rubber can be processed through crushing, cleaning and modification to make them suitable for HDC.

(2) With the addition of recycled or waste materials, it may be necessary to adjust the HDC mix ratio to ensure that the concrete's performance indicators still meet the design requirements.

(3) Through laboratory tests and field application verification, it is necessary to assess whether the mechanical properties (e.g., compressive strength, tensile strength, flexural strength, etc.), durability (e.g., impermeability, resistance to freezing and thawing, etc.), and workability (e.g., fluidity, setting time, etc.) of HDC are changed by the addition of recycled or discarded materials. These tests should ensure that the performance of HDC is not affected or is affected within acceptable limits when recycled or waste materials are added.

(4) Economic and environmental considerations also need to be taken into account when deciding whether or not to incorporate recycled or discarded materials into HDC. The utilization of recycled or waste materials can reduce the consumption of natural resources, lower production costs, and help reduce construction waste and environmental pollution. However, attention also needs to be paid to the costs of obtaining, handling and transporting these materials and their possible impact on HDC productivity and construction speed.]

Comments 5: [Please provide more details in text about Figure 1.]

Response 5: Agree. We have completed the revisions to emphasize this point. We have provided more details about Figure 1 in the main text. This change can be found in the revised manuscript – page 3, Paragraph 1, and lines 107-110.]

Comments 6: [Please change the resolution of Figure 2.]

Response 6: Agree. We have completed the revisions to emphasize this point. We have changed the resolution of Figure 2. This change can be found in the revised manuscript – page 5, Paragraph 2, and lines 186.]

Comments 7: [Please change the resolution of Figure 4.]

Response 7: Agree. We have completed the revisions to emphasize this point. We have changed the resolution of Figure 4. This change can be found in the revised manuscript – page 4, Paragraph 1, and line 266.]

Comments 8: [Please refer to the strategies that can be employed to enhance modeling and simulation tools for more precise predictions of HDC behavior and improved mix design optimization.]

Response 8: [I am very sorry to say that this paper is a review research paper and is not applicable to the strategy of enhancing modeling and simulation tools to more accurately predict HDC behavior and improve hybrid design optimization, so this strategy cannot be adopted.]

Round 2

Reviewer 1 Report

Comments and Suggestions for Authors

In the actual version, in general, all suggestions given by the reviewer was commented.

Comments on the Quality of English Language

Minor editing of English language required.

Author Response

Comments 1: In the actual version, in general, all suggestions given by the reviewer was commented.

Response 1: Agree. We have annotated all the suggestions you have given in red.

  1. Response to Comments on the Quality of English Language

Point 1: We have made a small number of edits to the English of the article.

Response 1: For more details, please check the main text.

Reviewer 2 Report

Comments and Suggestions for Authors

The authors have answered my questions. However, I have one last comment: while the use of references is essential to support a paper and demonstrate the relevance of existing research, an excess of citations can be problematic. In the case of this article, an excessive number of references is observed, which may negatively impact its quality in various aspects. A critical review of the references used is advisable, prioritizing those that are truly essential for constructing the central arguments of the article. Focusing on key studies and reducing redundant or marginally relevant citations could enhance the clarity and originality of the work, as well as facilitate readability and comprehension for the target audience.

Comments on the Quality of English Language

Minor editing of English language required.

Author Response

Comments 1: The authors have answered my questions. However, I have one last comment: while the use of references is essential to support a paper and demonstrate the relevance of existing research, an excess of citations can be problematic. In the case of this article, an excessive number of references is observed, which may negatively impact its quality in various aspects. A critical review of the references used is advisable, prioritizing those that are truly essential for constructing the central arguments of the article. Focusing on key studies and reducing redundant or marginally relevant citations could enhance the clarity and originality of the work, as well as facilitate readability and comprehension for the target audience.

Response 1: Agree. We have changed the number of references from 170 to 125 as you requested by focusing on key research and reducing redundant or marginally relevant citations.

  1. Response to Comments on the Quality of English Language

Point 1: We have made improvements to the quality of the English language.

Response 1: For more details, please check the main text.

Reviewer 3 Report

Comments and Suggestions for Authors

This paper can be published in its present form. 

Author Response

Comments 1: This paper can be published in its present form.

Response 1: Agree. We appreciate your comments and suggestions.
